# Validation and Psychometric Properties of the German Operational and Organizational Police Stress Questionnaires

**DOI:** 10.3390/ijerph20196831

**Published:** 2023-09-26

**Authors:** Lauriane Willemin-Petignat, Royce Anders, Sabrina Ogi, Benjamin Putois

**Affiliations:** 1Faculty of Psychology, UniDistance Suisse, 3900 Brig, Switzerland; 2Psychological Service, Bern Police Department, 3001 Bern, Switzerland; 3Department of Psychology, Epsylon Laboratory UR4556, University Paul Valéry Montpellier 3, 34000 Montpellier, France; 4Faculty of Applied Faculty, University of Applied Sciences and Arts Northwestern Switzerland, 4600 Olten, Switzerland; 5Lyon Neuroscience Research Center, National Centre for Scientific Research UMR 5292—National Institute of Health and Medical Research U1028, 69675 Bron, France

**Keywords:** psychometric validation, operational stressors and organizational stressors, Police Stress Questionnaire (PSQ), police officers

## Abstract

Context: Working in the police force is an operationally and organizationally stressful job. Suitable psychometric tools are needed to detect and study the psychosocial risks of these professionals. The original version contains 40 items, which may be too long for clinical use or as a research control measure. The main aim of this study is to validate the Police Stress Questionnaire (PSQ) in German. The secondary objective is to validate a shorter version. Method: After translation and counter translation of the PSQ-G by a committee of experts, 10 participants pre-tested the comprehension of an intermediate version, allowing the development of a final version that was submitted to a psychometric validation plan with 2314 German-speaking officers. Structure, reliability, and convergent, divergent, and discriminant validities were tested for each sample. Results: The German version of the PSQ performed well psychometrically. We have created a short version of 14 items with good psychometric properties, 7 items for each subscale: operational stressors and organizational stressors. Conclusion: This study validated a German version of the PSQ and provides a reliable measure of stress processes in the police force. A short version is now available.

## 1. Introduction

Policing is well known to be a highly stressful profession [1,2,3]. Officers are routinely confronted with a multitude of possible stressors [4]. In the literature e.g., [5], these stressors have been organized into two main categories: operational stressors, related to job responsibilities and performance, and organizational stressors, related to the managerial and logistical settings to which the officer is exposed.

Typical examples of operational stress include the stress associated with assuring one’s duties during shift work, being confronted with public aggression and noncompliance, being exposed to traumatic events such as witnessing grotesque outcomes of accidents and crime scenes, being obliged to use one’s firearm, being injured in the line of duty, and balancing work demands with private life. In contrast, typical examples of organizational stress include the stress associated with insufficient material and structural resources to adequately or comfortably perform one’s job, the lack of human resources for correct job outcomes, difficult relationships with colleagues and supervisors, having to align to excessive administrative tasks or bureaucracy, and too frequent changes in laws and procedures that may lead to confusion or frustration [2]. Various studies have corroborated that higher, and especially chronic rates of these stressors are not only associated with higher rates of burnout, post-traumatic stress disorders, anxiety, or depression are observed [6,7,8], but also with increased risks of cardiovascular disease [9] and the development of metabolic syndrome [10]. In addition, several studies have shown that high levels of stress can significantly affect the performance of police officers. Indeed, although general performance can be improved, a deterioration in verbal communication [11] and proportionality decision errors [12] have also been observed.

Several tools have been developed over time to measure occupational stress [13,14,15,16]. However, these measures do not take into account the stressors specific to police officers. In addition, McCreary and Thompson [5] point out that other questionnaires specific to the police profession have not been scientifically validated e.g., Police Stress Survey [PSS], [17], are too long e.g., Police Daily Hassles Scale, [18] or do not take into account work–life balance.

In aims to accurately measure the degree of these two sources of stress in police officers, McCreary and Thompson [5] developed a formal questionnaire specially for this population, known as the Police Stress Questionnaire (PSQ), which is generally known as the standard today. Indeed, the PSQ is divided into two subscales, each containing 20 items, that respectively measure operational stress (PSQ-Op) and organizational stress (PSQ-Org). While the PSQ has been translated into many languages, only several (i.e., Italian, Portuguese, Serbian, and Malay versions) have been formally evaluated for their cross-cultural integrity and validated psychometrically in the form of peer-reviewed articles [19,20,21,22]. These four peer-reviewed translations demonstrated appropriate reliability and validity of the scale, for example, with strong Cronbach’s alphas ranging from 0.93 to 0.96.

The present article sets forth to develop two primary contributions for the existent literature, for which the second extends beyond a single language. That is, the first aim of the present study was to appropriately adapt the PSQ to the German language, and, following the canonically recommended scale validation steps in the literature, evaluate its psychometric validity with a robustly large sample of German-speaking police officers from Switzerland. The second aim was motivated by the interest in developing a shorter PSQ version that may be advantageous for higher dimensional, multivariate studies including police stress, for which total questionnaire length may encourage better response rates e.g., [23] and hence statistical power. Therefore, the second aim of this study was to develop an abridged PSQ version, based on exploratory factor analysis goodness-of-fit measures (e.g., factor loadings, item uniqueness) as well as those of confirmatory factor analysis (e.g., coefficients and satisfactory model fit indices). The retained items of the short PSQ may be considered in its adaptation to other languages.

## 2. Methodology

### 2.1. Procedures

Following modern recommended methods in the scientific literature [24] and also taking into account those by the World Health Organization [25], accordingly, the present work was conducted in three stages: 1. Translating the PSQ to a tentative German version, as well as its back-translation to English; 2. Pre-testing the preliminary version, adapting where necessary, and converging on a final version; and then 3. Evaluating and ultimately validating the psychometric adequacy of the German PSQ based on its appropriate fulfillment of the performance benchmarks in the expected statistical analyses.

All participants gave their consent prior to participation, and all participation was anonymous. This research was validated by the relevant Swiss research ethics committee, known as “Zentrum für Ausbildung der Hochschule für Angewandte Psychologie” (FHNW), reference no. EAL22030.
−Phase 1: Translation into German and back-translation

The PSQ was translated and back-translated by experienced psychological staff working in the police department who possessed an advanced level of both German and English. The forward translation was performed by Ariane Delessert, licensed psychologist, and assistant Sylvia Monneron, both working for the Bern Police Department. It was then back-translated by Lauriane Willemin-Petignat, licensed psychologist and police officer, and Ariane Delesssert, licensed psychologist also working for the Bern Police Department. This work culminated in a preliminary version which was then subjected to the recommended standard questionnaire comprehension tests.
−Phase 2: Pre-testing, assessment of item comprehension, and final version development

The preliminary version of the German PSQ was provided in written form to 10 officers working for the Bern Police Department. Each officer noted their understanding of each item from 1 (“this expression is very unclear”) to 7 (“this expression is very clear”), and they were also allowed to propose modifications. For each question, the mean scores were calculated, and all items reached a mean score of 5 or higher. Any modification proposals for specific items were meticulously considered by the previously mentioned experts and were adapted where deemed a true improvement was needed.
−Phase 3: Evaluation of the translated scale’s measurement validity

The final German version of the PSQ, hereafter the PSQ-G, was then evaluated using the canonical analyses recommended for evaluating its adequacy as a measurement tool. These include individual item analyses, calculating and evaluating the internal consistency measures Cronbach’s alpha and Donald’s omega of the full scale, and the two subscales, exploratory factor analyses and confirmatory factor analyses. These results are presented after the Methods. 

### 2.2. Participants

All participants were police officers who, at the time of data collection, possessed a Swiss Federal Patent (or equivalent, for the older participants) and were working for the police forces of the German-speaking regions within Switzerland. The officers worked in a variety of suburban or municipal police forces and services.

### 2.3. Measures

Participants filled out a panel of four questionnaires through the secure platform of the FHNW. It was followed that, according to the FHNW ethics committee, none of the questions of the survey should be mandatory, as some of them were related to perceived trauma. In addition to the PSQ-G, we administered questionnaires on mental disorders that are typically associated with stress and/or are common among police officers [6], namely, post-traumatic stress disorder, in which stress persists over long periods following a traumatic event; burn-out syndrome, associated with the physiological and psychological expression of prolonged exposure to stressors; and anxiety and depression, which accentuate the perception of stress [26].

#### 2.3.1. Police Stress Questionnaire Adapted to German (PSQ-G)

The original English PSQ [5] consists of a 40-item self-report scale measuring stress in policing careers, organized into two subscales: operational stress (PSQ-Op) and organizational stress (PSQ-Org). For each item, the participants are asked how much stress they have had related to it over the past 6 months. Responses are given on a 7-point Likert scale ranging from 1 (“no stress at all”) to 7 (“a lot of stress”). Means on each scale are used to compare to a threshold indicating high stress in that domain, that is, according to McCreary et al. [27], a mean of 4.7 or greater on the PSQ-Op, or 4 or greater on the PSQ-Org.

#### 2.3.2. Impact of Event Scale—Revised (IES-R)

The IES-R [28] is a 22-item self-report scale measuring symptoms of post-traumatic stress disorder (PTSD), in which herein the German adaptation was used [29]. The scale is divided into 3 subscales: intrusion, avoidance, and hypervigilance. The participant is asked to identify a specific stressful event to them and indicate how much difficulty this event has caused them in the past 7 days in different respects. In the German version, responses are given on a 4-point Likert scale ranging from 0 (“not at all”) to 4 (“often”). PTSD positivity is determined by a nonlinear combination of the three subscales.

#### 2.3.3. Oldenburg Burnout Inventory (OLBI)

The OLBI [30] is a 16-item self-report scale operationalizing the dimensions of professional exhaustion and disengagement. The participant is asked to what extent he agrees with each of the statements. Responses are given on a 4-point Likert scale ranging from 1 (“strongly agree”) to 4 (“strongly disagree”). High burnout symptomology is identified with this scale via a score above 1 standard deviation of the mean [31].

#### 2.3.4. Hospital Anxiety and Depression Scale (HADS)

The HADS [32] is a 14-item self-report scale, consisting of two subscales that separately measure anxiety and depression. Herein, the German adaptation was used [33]. In relation to anxiety or depression, participants rate how often they experience different phenomena. Responses are given on a 4-point Likert scale ranging from 0 to 3, wherein the description (of 0 to 3) may vary according to the question. The typical clinical threshold suggesting the presence of anxiety or depression is a score ≥ 8 (in their respective subscales), and a total score ≥ 11 may reflect a general adjustment disorder related to anxiety or depression [34].

## 3. Data Analysis

All data collected were analyzed using Python 3.9. First, the following analyses were performed for the PSQ-G and then used to derive (in a data-driven manner) and evaluate a short version PSQ-G that resulted in 14 items (PSQ-G-14). 

### 3.1. Reliability

Reliability as per the internal consistency of the scale and subscales in the PSQ were assessed via the Cronbach’s alpha (α) and McDonald omega (ω) statistics. Values greater than 0.70 are considered to indicate appropriate internal consistency [35,36]. 

### 3.2. Construct Validity

The conventional exploratory factor analyses (EFA) and confirmatory factor analyses (CFA) for evaluating construct validity were performed for the PSQ-G and PSQ-G-14. 

#### 3.2.1. Exploratory Factor Analyses (EFA)

First, Bartlett’s test of sphericity significance preferred [37] and the Kaiser–Meyer–Olkin (KMO) measure of sampling adequacy KMO ≥ 0.60 preferred [38] were evaluated to assess the appropriateness of performing factor analysis on the data. For the EFA, scree plot analysis was then used to ascertain the number of factors, based on identifying at which factor the eigenvalues descend linearly and slowly suggesting mainly noise being thereafter fit, known as the “elbow” method [39]. For both the PSQ-G and PSQ-G-14, a two-factor solution was evident. Then, the EFA was performed using the minimal residual approach with the varimax rotation method [40]; the latter known as one of the most orthogonal methods. The factor loadings were then examined to verify that the items loaded appropriately onto the expected factors (reflecting the respective subscales, PSQ-Op and PSQ-Org). 

The short version of the PSQ, the PSQ-G-14, was primarily determined through a data-driven approach with the EFA, e.g., [41]. In a stepwise iterative process, the item with the weakest factor loading was removed up until all loadings were greater than or equal to 0.5. Then, the items with weak communality (<0.35) were removed, e.g., [42]. This resulted in 7 items remaining in the PSQ-Org, and 13 in PSQ-Op; the 7 items with the strongest factor loadings were retained in the PSQ-Op, resulting in 14 items total, 7 in each scale for the short PSQ-G. Details on which items were retained from the PSQ-G, their content, and the EFA results of the PSQ-G-14 are provided in Table 1.

#### 3.2.2. Confirmatory Factor Analyses (CFA)

Using the conventional CFA approaches that are recommended for this purpose [43], the latent factor structure of the PSQ-G and PSQ-G-14 were analyzed, and notably, with a robust sample (participants *N* = 2314). The two subscales of the PSQ were modeled according to their expected membership. The adequation of model fit was evaluated based on the adjustment indices defined by [44]: χ²/*df*, the Comparative Fit Index CFI, [45]; the Root Mean Squared Error of Approximation RMSEA, [46]; and the Standardized Root Mean Square Residual SRMR, [47]. The preferred values for these indices are as follows: (a) χ²/*df* < 5; (b) CFI > 0.90; (c) RMSEA < 0.10; (d) SRMR < 0.08 [48]. Then, the item coefficients and the covariance between factors were examined. Item coefficients above 0.5 are considered acceptable, and covariances above 0.4 that are significant (*p* < 0.01) are considered to make a reasonable contribution to the model [49]. Finally, the R2 values of each variable were analyzed to determine the reliability index of each observed variable as a measure of its latent variable. These values should be positive and tend toward 1 [50].

### 3.3. Convergent Validity

The convergent validities of the PSQ-G and PSQ-G-14 were evaluated by the examination of correlations between different measurement tools. Prior to calculating the Pearson’s correlation coefficients, which assume normality, the data were normalized via the Yeo–Johnson transformation [51]. Correlation coefficients ≥ 0.4 indicate a moderate correlation, which becomes strong when these coefficients are ≥0.7 [52]. The *p*-values of these correlations were corrected for multiple comparisons using the Bonferroni–Holm correction [53], and significance was accepted according to corrected *p*-values < 0.05.

## 4. Results

### 4.1. Phase 2: Pre-Test

After the translation and back translation procedures detailed in the Methods, the questionnaires were pretested: 10 German-speaking police officers evaluated a preliminary version of the PSQ in German. According to the mean scores obtained for each item (>5), all were appropriately understandable and were kept as is. On a Likert scale from 1 to 7 (7 being “very clear”), the mean score for comprehension across all items was 6.1. These participants did not provide their age or sex.

### 4.2. Phase 3: Psychometric Validation

#### Participants

Out of a total of 2587 participants, 273 were removed due to missing data. The participants in this study (*N* = 2314), represented a participation rate of 26% of the total population of German police officers in Swiss, constituting 24 different police forces. The sample is primarily male (79% men), with an average age of 41.9 years (±9.8) and 16.5 years of work experience (±42.6). In this sample, 10.8% of officers were single and 60.5% did not have children. Of the 2314 participants, 35.1% were working in emergency police, 24.2% in judicial police, 16.6% in community police, and 24.1% in other services. 

### 4.3. Factor Analyses

The conventional factor analyses for scale validation were performed on the 40 items of the PSQ-G (hereafter PSQ-G) and the 14 items of the PSQ-G-14, specifically exploratory factor analysis (EFA) and confirmatory factor analysis (CFA). First, the standard preliminary checks to ascertain the suitability of data for the factor analysis were performed. Specifically, the Kaiser–Meyer–Olkin Measure of Sampling Adequacy (KMO) test was satisfied with a KMO = 0.95 for the PSQ-G and KMO = 0.96 for the PSQ-G-14 (values ≥ 0.6 preferred) and the Bartlett’s test of sphericity value was also satisfied for both scales, each with *p* < 0.001 (*p* < 0.05 preferred). 

#### 4.3.1. Exploratory Factor Analysis (EFA)

After the scree plot verification of two principal factors, an EFA limited to two-factor solution was applied to PSQ-G. The two-factor solution explained 37.7% of the total variance, in which Factor 1 accounted for 20.4% and Factor 2 for 17.3%. All items exhibited higher factor loadings in their expected factor (subscale) than the alternative, except for PSQ-Op 9, PSQ-Org 10, and PSQ-Org 15. All item communalities were ≥0.2 (See Appendix A, for loading factors).

In respect to the PSQ-G-14, the two-factor solution robustly explained 53.2% of the total variance, in which Factor 1 accounted for 27.9% and Factor 2 for 25.3%. All items exhibited higher factor loadings in their expected factor (subscale) and all item communalities were ≥0.34.

#### 4.3.2. Confirmatory Factor Analysis (CFA)

As provided in Table 2, both the PSQ-G and PSQ-G-14 strongly satisfied six out of the seven indices of model adequacy, suggesting appropriate coherence of the latent factor structure. Furthermore, it was found that all of the standardized individual item coefficients were appropriately of magnitude 0.5 or greater. In each of the model fit indices, the PSQ-G-14 outperformed the PSQ-G. 

#### 4.3.3. Reliability: Internal Consistency

The PSQ-G demonstrated strong internal consistency for the total scale as well as for the two subscales: total scale Cronbach’s α = 0.95, Donald’s ω = 0.95; Op subscale α = 0.93, ω = 0.93; Org subscale α = 0.91, ω = 0.91. These results demonstrate strong internal consistency for the PSQ-G. 

The PSQ-G-14 also demonstrated strong internal consistency for the total scale as well as for the two subscales: total scale Cronbach’s α = 0.91, Donald’s ω = 0.93; Op subscale α = 0.90, ω = 0.89; Org subscale α = 0.87, ω = 0.89. 

#### 4.3.4. Convergent Validity

The left part of Table 3 provides the Pearson correlations of the PSQ-G and the other variables detailed in the Methods. As in Table 3, both subscales of the PSQ-G were significantly positively associated with measures of burnout, trauma, anxiety, and depression. The right part of Table 3 provides the same comparisons but with regard to the PSQ-G-14. The magnitude and significance of the PSQ-G-14 correlations are on par with the longer version of the scale, suggesting a similar degree of convergent validity. 

## 5. Discussion

The first aim of this study was to adapt the PSQ to the German language in accordance with modern guidelines e.g., [24], and evaluate its psychometric validity, e.g., [54]. The second aim was to develop a shorter version of the PSQ in order to facilitate its incorporation in multivariate studies where overall questionnaire length is often scrutinized. In regard to the police population sample recruited in light of these objectives, the notably large cohort (*N* = 2314) provided for robust statistical power. 

First, the internal consistency for the PSQ-G (classic 40 items) was found to be strong for the total scale, as well as for the two subscales (α = 0.95, ω = 0.95; Op subscale α = 0.93, ω = 0.93; Org subscale α = 0.91, ω = 0.91), making the PSQ-G-40 a reliable tool in the general population, in concordance with results found for the original English version [5]. Therefore, reliability of PSQ-G-40 was below the values reported by Setti [PSQ-Org: α = 0.95, PSQ-Op: α = 0.94, 22]; Queiros et al. [PSQ-Op: α = 0.96, 20]; Queiros et al. [PSQ-Org: α = 0.95, 21] and Kukic et al. [PSQ-Op: α = 0.96, PSQ-Org: α = 0.96, 19], but still considered good. Similar results were found for the PSQ-G-14 (total scale Cronbach’s α = 0.91, Donald’s ω = 0.93; Op subscale α = 0.90, ω = 0.89; Org subscale α = 0.87, ω = 0.89), which supports the use of the shorter scale as a practical alternative, when questionnaire length is of importance. 

Next, the confirmatory factor analyses herein demonstrated that the items of our two German versions respect an appropriate fit of a latent two-factor structure respectively for Operational and Organizational Stress. Specifically, six out of the seven standardly-reported indices were satisfied for the PSQ-G (χ2/df = 11.37; CFI = 0.96; IFI = 0.96; NNFI = 0.95; RMSEA = 0.07; GFI = 0.96; AGFI = 0.96), with an even better fit for the PSQ-G-14 (χ2/df = 10.67; CFI = 0.98; IFI = 0.98; NNFI = 0.98; RMSEA = 0.06; GFI = 0.99; AGFI = 0.98). Furthermore, the fit performance of the full version was comparable to that of recent Portuguese versions [20,21]. 

Thirdly, our translated PSQ-G and a statistically motivated proposition of an abridged 14-item version (PSQ-G-40) demonstrated good convergent validity. The total scores of each PSQ-G and their subscales correlated in the expected direction with burnout, trauma, anxiety, and depression, as suggested by authors of the original scale [5]. Considering the credibility of the large sample herein, it is a contribution to note that the PSQ-G was most strongly correlated with the exhaustion dimension of burnout, then anxiety and depression (similar scores), and finally, the disengagement dimension of burnout and hyperarousal of PTSD. This result encourages the coupling of a PTSD questionnaire as the other dimensions are the least correlated. For both the full and abridged PSQ versions, the total score was the most strongly correlated with other scales, suggesting that it may be a more effective predictor of these variables than the individual subscales. In regard to the subscales, the operational subscale was more strongly correlated than the organizational subscale. Some limitations of our study could be noted as criterion validity data were not available to examine with the usual *t*-test analyses. Moreover, divergent validity data were not available for comparison either. Nevertheless, the culmination of results herein reliably show the PSQ-G and PSQ-G-14 versions as satisfactory tools for the assessment of occupational stress in German police officers. 

Future studies assessing the measurement fidelity of such questionnaires would do well to also collect data on the psychiatric diagnoses associated with stress, such as post-traumatic stress disorder, depression, anxiety and burnout; leading to a more robust evaluation of convergent/divergent validity. In this same goal, experimental measures, e.g., the Trier Social Stress Test [55], or physiological ones (e.g., heart rate variability, electrodermal activity) could be used, leading to much more objective prowess than self-report questionnaires. Though, such a study design would likely result in a much smaller sample size than the present study.

## 6. Conclusions

This study found both scales of the developed German versions of the Police Stress Questionnaire to be valid and reliable instruments that enable professionals to assess perceived stressors among German-speaking police officers. In addition, being able to determine whether the stress experienced by police officers is caused by organizational or operational factors allows for better targeting of response measures that could be taken within police forces, allowing for a reduction in the inherent costs related to the mental health and well-being of officers.

Police officers comprise a population that is at greater risk of exposure to sporadic, intense stressors (assault, traffic accidents, and so forth), but they are also notably tasked with regulating both immediate and long-term stressful situations: conflict management, crisis management, and human management in emergency situations. Stress management is therefore particularly important for police officers. Can a police officer who is stressed by his or her work on an organizational or operational level properly fulfill his or her missions and duties? For example, shooting [56] performance under pressure or the decision criterion to shoot [57] may depend on the stress level and coping skills of the police officer. Taking care of the mental health of police officers is a highly relevant way of guaranteeing better interventions with the general population [58]. This questionnaire offers a simple and effective measurement tool, both clinically (for occupational psychologists and physicians) and for researchers, to understand and reduce these risks.

## Figures and Tables

**Table 1 ijerph-20-06831-t001:** Retained Items of the PSQ-G-14 and their EFA statistics.

Items		Operational	Organizational	Communalities
PSQ-G-Op-7	Managing your social life outside of work Management des soziales Lebens ausserhalb der Arbeit	0.74	-	0.57
PSQ-G-Op-8	Not enough time available to spend with friends and family Zu wenig Zeit für Freunde und Familie	0.82	-	0.72
PSQ-G-Op-11	Finding time to stay in good physical condition Die Zeit finden, um in guter körperlicher Verfassung zu bleiben	0.66	-	0.48
PSQ-G-Op-12	Fatigue (e.g., shift work, over-time) Ermüdung (z. B. Schichtarbeit, Überstunden)	0.67	-	0.55
PSQ-G-Op-18	Limitations to your social life (e.g., who your friends are, where you socialize) Einschränkung im sozialen Leben (z. B. Wer Ihre Freunde sind, wo Sie sich treffen)	0.69	-	0.51
PSQ-G-Op-19	Feeling like you are always on the job Das Gefühl, ständig im Einsatz zu sein	0.67	-	0.55
PSQ-G-Op-20	Friends/family feel the effects of the stigma associated with your job Freunde/Familie spüren die Auswirkung des Berufsstigmas	0.66	-	0.52
PSQ-G-Org-4	Excessive administrative duties Übermässige administrative Aufgaben	-	0.76	0.6
PSQ-G-Org-5	Constant changes in policy/legislation Ständige Änderungen in der Politik/Gesetzgebung	-	0.56	0.34
PSQ-G-Org-6	Staff shortages Personalmangel	-	0.66	0.53
PSQ-G-Org-7	Bureaucratic red tape Bürokratischer Aufwand	-	0.85	0.75
PSQ-G-Org-8	Too much computer work Zu viel Computerarbeit	-	0.63	0.43
PSQ-G-Org-13	Lack of resources Ressourcenmangel	-	0.64	0.52
PSQ-G-Org-14	Unequal sharing of work responsibilities Ungleiche Aufteilung der Arbeitsaufgaben	-	0.53	0.38
SS Loadings		3.91	3.54	
Explained Variance		27.9%	25.3%	

Note: *N* = 2314; loadings less than 0.35 are indicated in the table as “-”.

**Table 2 ijerph-20-06831-t002:** Confirmatory factor analysis (CFA) results for the PSQ-G full version and the PSQ-G-14 item version.

Indices	PSQ-G Observed Value	PSQ-G-14 Item Version Observed Value	Acceptable Threshold
Model χ^2^/*df*	11.37	10.67	<5.0
CFI ^1^	0.96	0.98	>0.90
IFI ^2^	0.96	0.98	>0.90
NNFI ^3^	0.95	0.98	>0.90
RMSEA ^4^	0.07	0.06	<0.10
GFI ^5^	0.96	0.99	>0.90
AGFI ^6^	0.96	0.98	>0.90

Note: ^1^ Comparative Fit Index; ^2^ Boellen’s Incremental Fit Index; ^3^ Non-Normed Fit Index (also known as Tucker–Lewis Fit Index); ^4^ Root Mean Square Error of Approximation; ^5^ Goodness-of-Fit Index; ^6^ Adjusted Goodness-of-Fit Index.

**Table 3 ijerph-20-06831-t003:** Pearson correlations between PSQ-G items and burnout, posttraumatic stress, anxiety, and depression variables (n = 2314), and likewise for the PSQ-G-14 item version.

	PSQ-G			PSQ-G-14 Item Version		
Dimensions	Total	Oper	Orga	Total	Oper	Orga
PSQ-G Total						
PSQ-G Operational	0.93 ***			0.88 ***		
PSQ-G Organisational	0.93 ***	0.73 ***		0.88 ***	0.55 ***	
OLBI Disengagement	0.44 ***	0.39 ***	0.42 ***	0.40 ***	0.36 ***	0.35 ***
OLBI Exhaustion	0.63 ***	0.61 ***	0.56 ***	0.64 ***	0.61 ***	0.51 ***
IES-R Hyperarousal	0.44 ***	0.41 ***	0.40 ***	0.39 ***	0.37 ***	0.32 ***
IES-R Intrusion	0.37 ***	0.36 ***	0.33 ***	0.32 ***	0.31 ***	0.26 ***
IES-R Avoidance	0.38 ***	0.35 ***	0.34 ***	0.33 ***	0.31 ***	0.28 ***
HADS Anxiety	0.50 ***	0.48 ***	0.44 ***	0.46 ***	0.47 ***	0.35 ***
HADS Depression	0.50 ***	0.47 ***	0.46 ***	0.49 ***	0.48 ***	0.37 ***

Note: *** *p* ≤ 0.001, all *p*-values corrected for multiple comparisons by the Holm–Bonferroni correction.

## Data Availability

No data are available.

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
