# Peer review of "Validation and Psychometric Properties of the German Operational and Organizational Police Stress Questionnaires"

_ijerph, 2023, doi:10.3390/ijerph20196831_

Round 1

Reviewer 1 Report

Dear authors, thank you for the opportunity to read your research.

Validation of the German version of the questionnaire was carried out at a high methodological level.

The introduction demonstrates the relevance of the study, and also provides information on the development of the questionnaire, contains the purpose of the study.

Further, the stages of questionnaire validation are clearly described, the research methods are described. The authors selected adequate methods for statistical data analysis.

The results are described in detail, structured and visually.

At the same time, there are small recommendations for improving the quality of the manuscript:

1. The authors did not indicate in the methods, in connection with which these particular methods (IES-R, OLBI, HADS) were chosen to assess the validity of the questionnaire?

2. The results of their relationship are discussed very little. It could be clarified with regard to stress levels and other assessed characteristics in this sample in order to illustrate the results, to show their importance and interest for practical purposes. This would expand the discussion of the results, as well as outline practical recommendations. The study sample is very large and, undoubtedly, the results obtained on the severity of operational and organizational stress of police officers would be very interesting for readers to correlate with other data.

3. Discussion of results is very limited, no analysis with other studies. Only a conclusion is made regarding the tested hypotheses. This section needs to be expanded.

With respect for your work and best wishes, reviewer

Reviewer 2 Report

The authors propose the adaptation of the short version of a scale for the assessment of stress in a high risk bournout population. The work appears well structured, the introduction seems clear and coherent and comprehensive. The hypothesis are adequately stated. The procedure appears well described, complete and in line with scientific protocols regarding the validation of psychometric instruments. The sample is very broad and the competing measures seem appropriate to the research objectives.

I ask the authors to:

- include in the introduction some data with respect to medical counselling requests in the military. For example, do you have any information on how many police officers seek medical advice in your country for stress-related problems?
- include some research that can show how stress can have an impact on the health of the operator (policeman) and, if possible, what impact it has on work performance.
- enrich the part on limits, which seems only hinted at. Some limitations may also concern the instrument itself and the other instruments used as self-report instruments.
- Enrich the practical applications and indications for future research.

I really appreciated the authors' work. I therefore suggest a minor revision.
